# Functional identification of soluble uric acid as an endogenous inhibitor of CD38

**Shijie Wen[1], Hiroshi Arakawa[1]\*, Shigeru Yokoyama[2,3], Yoshiyuki Shirasaka[1], Haruhiro Higashida[2], Ikumi Tamai[1]\***

[1]Faculty of Pharmaceutical Sciences, Institute of Medical, Pharmaceutical and Health Sciences, Kanazawa University, Kanazawa, Japan; [2]Research Center for Child Mental Development, Kanazawa University, Kanazawa, Japan; [3]Division of Socio-Cognitive-Neuroscience, United Graduate School of Child Development, Osaka University, Kanazawa University, Hamamatsu University School of Medicine, Chiba University and University of Fukui, Kanazawa, Japan

## eLife Assessment

This **important** study shows that soluble uric acid is an endogenous inhibitor of CD38, a regulator of inflammatory responses. The **convincing** evidence draws both on biochemical analyses and in vivo models. This work provides insights into NAD+ metabolism, with significant implications for inflammation and potential roles in metabolic diseases and aging.

**\*For correspondence:**
arakawa0403@gmail.com (HA);
tamai@p.kanazawa-u.ac.jp (IT)

**Competing interest:** The authors declare that no competing interests exist.

**Abstract** Excessive elevation or reduction of soluble uric acid (sUA) levels has been linked to some of pathological states, raising another subject that sUA at physiological levels may be essential for the maintenance of health. Yet, the fundamental physiological functions and molecular targets of sUA remain largely unknown. Using enzyme assays and in vitro and in vivo metabolic assays, we demonstrate that sUA directly inhibits the hydrolase and cyclase activities of CD38 via a reversible non-competitive mechanism, thereby limiting nicotinamide adenine dinucleotide (NAD$^+$) degradation. CD38 inhibition is restricted to sUA in purine metabolism, and a structural comparison using methyl analogs of sUA such as caffeine metabolites shows that 1,3-dihydroimidazol-2-one is the main functional group. Moreover, sUA at physiological levels prevents crude lipopolysaccharide (cLPS)-induced systemic inflammation and monosodium urate (MSU) crystal-induced peritonitis in mice by interacting with CD38. Together, this study unveils an unexpected physiological role for sUA in controlling NAD$^+$ availability and innate immunity through CD38 inhibition, providing a new perspective on sUA homeostasis and purine metabolism.

## Introduction

The evolutionary loss of uricase activity in humans and certain primates completely blocks the degradation of soluble uric acid (sUA), leading to higher physiological levels of sUA than in other mammals (*Oda et al., 2002*; *Wu et al., 1992*). Owing to renal reabsorption, sUA is strictly maintained in humans rather than being eliminated as a waste. It has been suggested that sUA functions as an abundant antioxidant (*Glantzounis et al., 2005*) and is crucial to maintain blood pressure (*Watanabe et al., 2002*). Although hyperuricemia may promote the precipitation of monosodium urate (MSU) crystal, an activator of the NLRP3 inflammasome (*Martinon et al., 2006*), resulting in diseases such as gout (*Dehlin et al., 2020*) and kidney stone disease (*Howles and Thakker, 2020*), numerous studies have indicated the protective potential of sUA in neurodegenerative diseases (*Kutzing and Firestein, 2008*; *Lu et al., 2016*; *Scott et al., 2002*). In addition, abnormal reduction of sUA levels is clinically

associated with the risk of various diseases (*Crawley et al., 2022*; *Kutzing and Firestein, 2008*), including cardiovascular diseases and kidney diseases. Intriguingly, rapid urate reduction in the initiation of therapy even increases gout flares in patients (*Becker et al., 2005*). Depletion of sUA in uricase-transgenic mice shortens lifespan and promotes sterile inflammation induced by microbial molecules or pro-inflammatory particles (*Kono et al., 2010*; *Shi, 2010*). Thus, data supports that sUA provides a physiological defense against excessive inflammation, aging, and certain diseases (*Álvarez-Lario and Macarrón-Vicente, 2010*; *Ames et al., 1981*; *Crawley et al., 2022*; *Cutler et al., 2019*; *Kutzing and Firestein, 2008*; *Lai et al., 2017*; *Linnerz et al., 2022*; *Ma et al., 2020*; *Wan et al., 2020*; *Wen et al., 2024*). However, the underlying molecular basis of sUA physiology remains poorly understood.

To gain insight into target-based physiological functions of sUA, our laboratory utilized magnetic bead-conjugated 8-oxoguanine (8-OG, an sUA analog; technically, sUA cannot be directly conjugated to the beads) and proteomic analysis to screen the potential candidates of sUA-binding proteins. We discovered several binding proteins of 8-OG (unpublished data), including CD38 that was verified by enzyme inhibition (*Figure 1A*), which raised our interest in exploring the role of CD38 in sUA physiology. CD38 is mainly expressed in immune cells and has type II or type III membrane orientation, with the catalytic domain facing the outside or inside of cells (*Hogan et al., 2019*; *Lee and Zhao, 2019*; *Zhao et al., 2012*). It is also found in intracellular membranes or as intracellular and extracellular soluble forms (*Chini et al., 2018*; *Funaro et al., 1996*; *Lee and Zhao, 2019*; *Zielinska et al., 2004*). Functionally, CD38 serves as a hydrolase to degrade nicotinamide adenine dinucleotide (NAD$^+$) (*Hogan et al., 2019*), an essential cofactor for various metabolic reactions that sustain life (*Luongo et al., 2020*), as well as its precursor nicotinamide mononucleotide (NMN) (*Chini et al., 2020*), thus regulating inflammation, aging, and various diseases (*Chini et al., 2018*; *Hogan et al., 2019*). Its cyclase catalyzes the synthesis of cyclic ADP-ribose (cADPR) from NAD$^+$ to drive calcium mobilization (*Lee, 2001*; *Lee et al., 1989*), which is crucial for social behavior (*Jin et al., 2007*) and neutrophil recruitment (*Partida-Sánchez et al., 2001*).

Here, we demonstrate that sUA at physiological levels directly inhibits CD38 and consequently limits NAD$^+$ degradation and excessive inflammation, which defines, for the first time, the physiological functions of sUA via CD38. In addition, we confirm the unique effect of sUA on CD38 in purine metabolism and identify a structural feature for pharmacological inhibition of CD38.

## Results

### sUA is an endogenous, reversible, and non-competitive inhibitor of CD38

To clarify whether CD38 is a direct target for sUA, we investigated the effect of sUA on CD38 activity. We found that sUA directly inhibited the hydrolase and cyclase activities of human (*Figure 1B*; *Figure 1—figure supplement 1A*) and murine (*Figure 1E*) CD38 as a non-competitive inhibitor (*Figure 1C and D*; *Figure 1—figure supplement 1B*) with a K$_i$ in the micromolar range (57.1–93.3 µM), demonstrating its binding to the allosteric sites of CD38. sUA showed comparable inhibitory effects on hydrolase and cyclase, as indicated by similar K$_i$. The physiological levels of sUA in humans (about 120–420 µM) (*Dalbeth et al., 2021*; *Kuwabara et al., 2017*) and in several mouse tissues (*Supplementary file 1*) were higher than its K$_i$, indicating that human CD38 and murine type III/intracellular CD38 are physiologically inhibited by sUA. The inhibitory effects were reversible (*Figure 1F and G*; *Figure 1—figure supplement 1C and D*), suggesting that sUA dynamically modulates CD38 activity. Interference from endogenous sUA was negligible when using tissues as an enzyme source, because the concentrations in the final reaction buffer were below 1 µM (*Figure 1—figure supplement 2*).

### CD38 inhibition is restricted to sUA in purine metabolism

Although the structure of sUA is similar to that of other purines, we confirmed its unique effect on CD38 in the major metabolic pathways of purines (*Figure 2A*). sUA precursors (adenosine, guanosine, inosine, hypoxanthine, and xanthine) and the uricase-catalyzed metabolite (allantoin) hardly inhibited the hydrolase and cyclase activities of human (*Figure 2B and C*; *Figure 2—figure supplement 1A*) and murine (*Figure 2—figure supplement 1B and C*) CD38. The tested purine concentrations were higher than their physiological levels (*Boulieu et al., 1983*; *Dudzinska et al., 2010*; *Eells and*

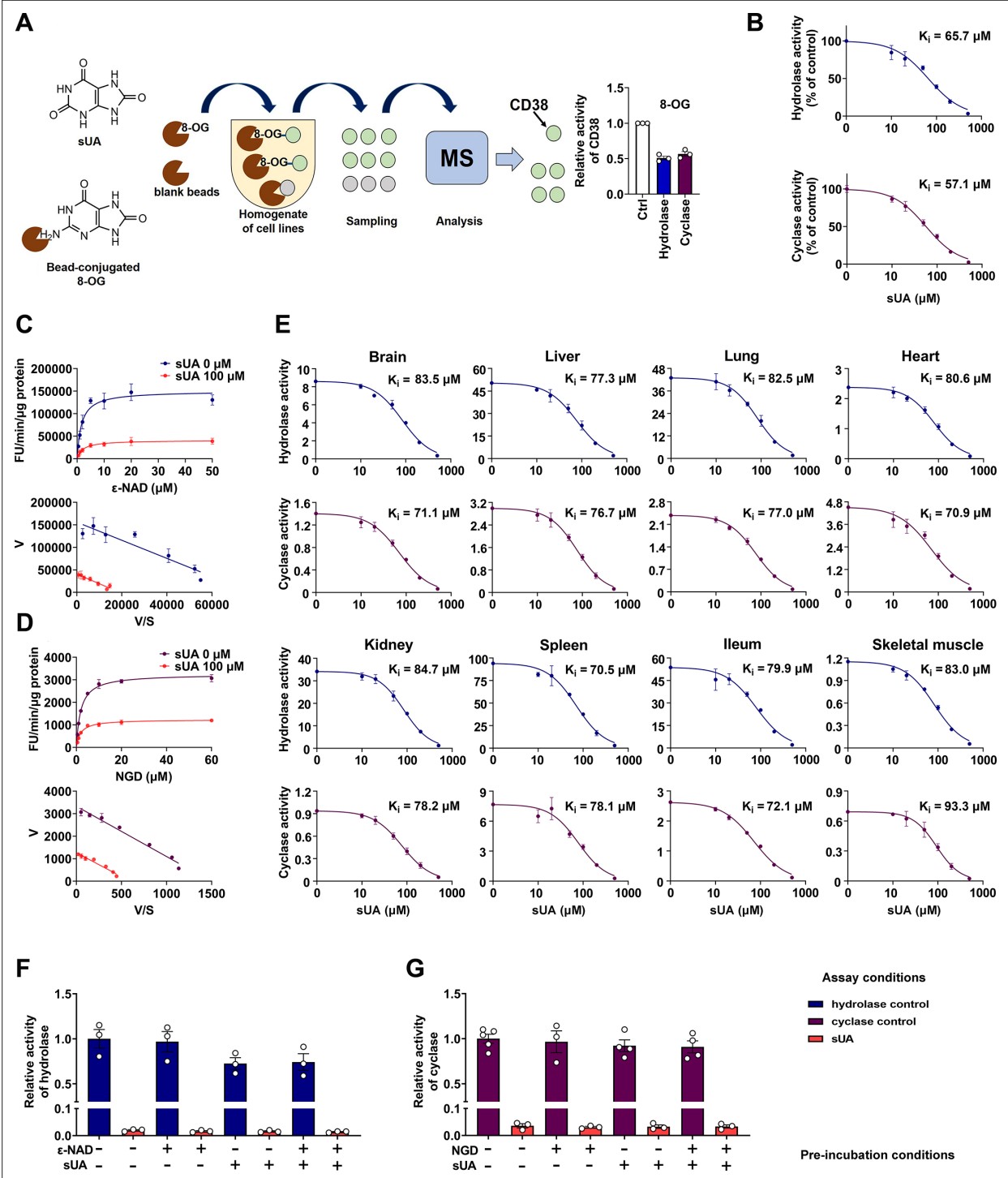

**Figure 1.** Identification of soluble uric acid (sUA) as an endogenous inhibitor for CD38. (**A**) Preliminary screening of 8-oxoguanine (8-OG) binding proteins by mass spectrum (MS)-based proteomics, and effect of 8-OG (50 μM) on CD38 activity (n=3 experiments/technical replicates). (**B**) Hydrolase and cyclase activities of recombinant human CD38 (hCD38) in the presence of sUA (0–500 μM), using nicotinamide 1, $N^6$-ethenoadenine dinucleotide (ε-NAD$^+$), and nicotinamide guanine dinucleotide (NGD) as substrates, respectively (n=3 experiments/technical replicates). (**C and D**) Effect of different substrate concentrations on sUA inhibition of recombinant hCD38 hydrolase (**C**) and cyclase (**D**) activities (n=3 experiments/technical replicates). (**E**) Effect of different sUA concentrations (0–500 μM) on hydrolase and cyclase activities (FU/min/μg protein) in tissues from 8- to 12-week-old wild-type (WT) mice (n=3 experiments from 3 mice). (**F and G**) Reversibility of inhibition of recombinant hCD38 hydrolase (**F**) and cyclase (**G**) activities by sUA. After 30 min pre-incubation as indicated, samples were diluted 100-fold in reaction buffer with or without 500 μM sUA for enzyme assay (n=3–5 experiments/ technical replicates). Data are mean ± s.d. (**B–E**) or mean ± s.e.m. (**A**, **F**, and **G**).

*Figure 1 continued on next page*

*Figure 1 continued*

The online version of this article includes the following figure supplement(s) for figure 1:

**Figure supplement 1.** Soluble uric acid (sUA) inhibition of CD38 and reversibility in THP-1 and A549 cells.

**Figure supplement 2.** Endogenous soluble uric acid (sUA) concentrations in the final reaction buffer for enzyme assays.

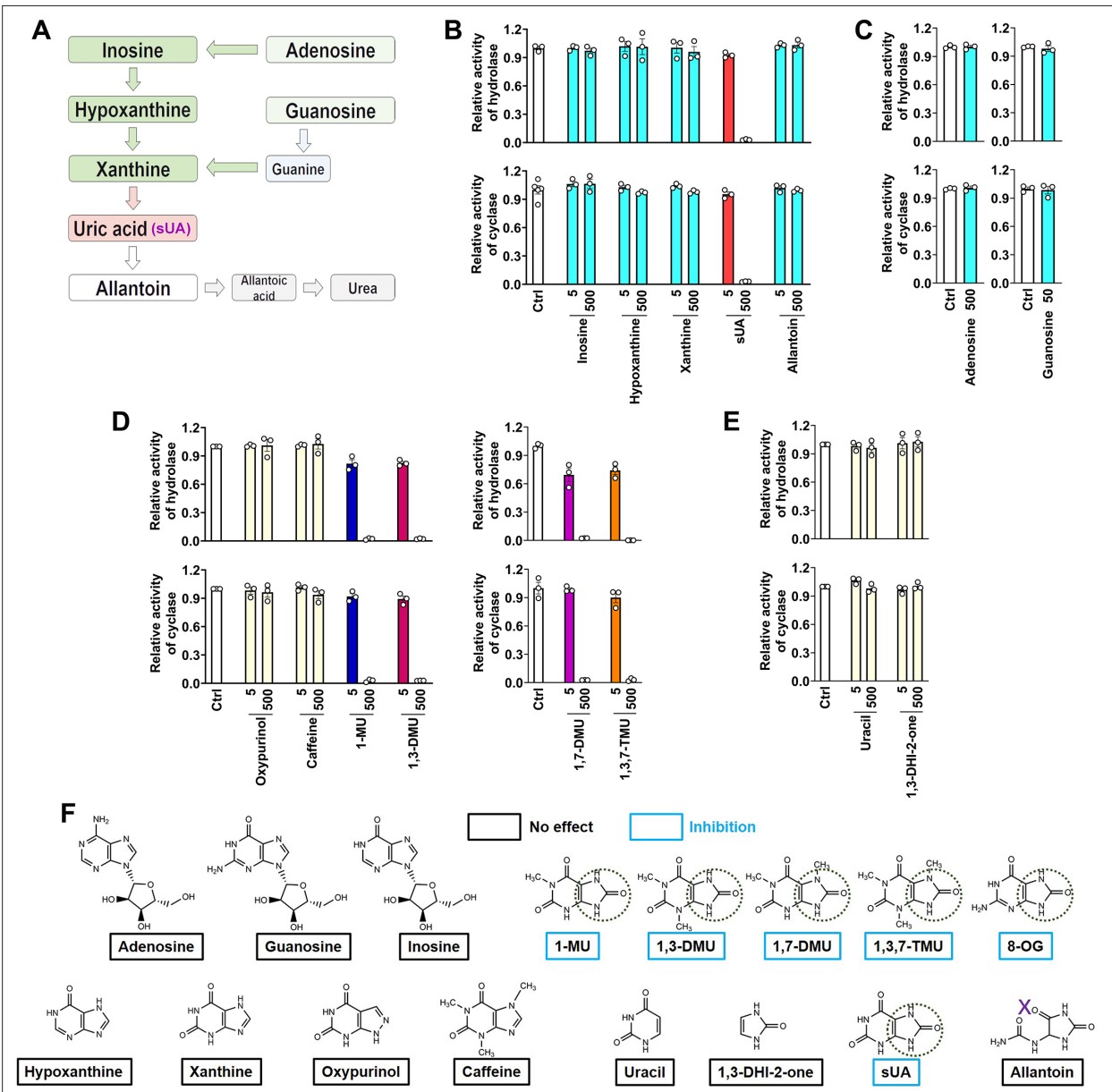

**Figure 2.** CD38 inhibition is restricted to soluble uric acid (sUA) in purine metabolism. (**A**) Major pathways of purine metabolism. (**B and C**) Effect of sUA and its precursors and metabolite on hydrolase and cyclase activities of recombinant hCD38 (n=3 experiments/technical replicates for each ligand). (**D**) Effect of different analogs on hydrolase and cyclase activities (n=3 experiments/technical replicates). THP-1 cells were used to detect the effects of oxypurinol, caffeine, 1-methyluric acid (1-MU), and 1,3-dimethyluric acid (1,3-DMU) on hydrolase activity, recombinant hCD38 was used in the remaining detections. (**E**) Effect of uracil and 1,3-dihydroimidazol-2-one (1,3-DHI-2-one) on hydrolase and cyclase activities of recombinant hCD38 (n=3 experiments/technical replicates). (**F**) A structural comparison reveals the functional group for CD38 inhibition. The concentrations of all ligands are from 5 to 500 μM. Data are mean ± s.e.m.

The online version of this article includes the following figure supplement(s) for figure 2:

**Figure supplement 1.** Effect of soluble uric acid (sUA), sUA precursors, and metabolite, and other derivates on CD38 activity in cells and tissues.

*Spector, 1983*; *Traut, 1994*). Thus, CD38 inhibition is restricted to sUA, suggesting a specific functional group in sUA.

To identify the functional group for CD38 inhibition, we tested additional xanthine analogs (oxypurinol and caffeine) and the methyl analogs (caffeine metabolites) of sUA including 1-methyluric acid (1-MU), 1,3-dimethyluric acid (1,3-DMU), 1,7-dimethyluric acid (1,7-DMU), and 1,3,7-trimethyluric acid (1,3,7-TMU). The results showed that only sUA analogs inhibited CD38 activity (*Figure 2D*). A structural comparison (*Figure 2F*) indicated that 1,3-dihydroimidazol-2-one (1,3-DHI-2-one) is the main functional group, as all other purines and derivates lacking this group failed to inhibit CD38 activity. In addition, the ring-opening of the uracil group after sUA conversion to allantoin abrogated this inhibitory potential. Uracil or 1,3-DHI-2-one (*Figure 2E*; *Figure 2—figure supplement 1D and E*) alone did not affect CD38 activity. Therefore, the adjacent uracil-like heterocycles are also essential for CD38 inhibition.

## sUA at physiological levels limits NAD$^+$ degradation by directly inhibiting CD38

Next, we explored the effect of sUA on NAD$^+$ availability, as CD38 is a key enzyme in degrading NAD$^+$ and its precursor NMN (*Chini et al., 2020*). sUA boosted intracellular NAD$^+$ in A549 and THP-1 cells (*Figure 3—figure supplement 1*) without affecting the activities of nicotinamide phosphoribosyltransferase (NAMPT) and poly(ADP-ribose) polymerase (PARP) (*Figure 3—figure supplement 2A and C*), two key enzymes involved in NAD$^+$ synthesis and metabolism, which suggests CD38 as a main target of sUA in regulating NAD$^+$ availability. Given that the physiological concentrations of hypoxanthine and xanthine are generally within 20 µM, both NAMPT and PARP might not be affected by purine metabolism under physiological conditions (*Figure 3—figure supplement 2B and D*). Thus, we further investigated whether CD38 mediates the effect of sUA on NAD$^+$ degradation. Short-term and moderate 'sUA-supplementation' model was constructed by gavage of inosine and oxonic acid (OA, a uricase inhibitor with a $K_i$ in the nanomolar range; *Fridovich, 1965*) in mice with 'natural hypouricemia' (*Figure 3A*; *Figure 3—figure supplement 5C*). The plasma sUA levels (around 120 µM) in our models were close to the minimum physiological concentrations in humans but were markedly lower than that in other long-term and hyperuricemia-associated disease models in rodents, which enabled us to evaluate the physiologically inhibitory effect of sUA on CD38 activity. Although OA was a weak inhibitor of CD38 with an IC$_{50}$ in the millimolar range (*Figure 3—figure supplement 3A and B*), it seemed unlikely to affect CD38 activity in our models because of the incomplete inhibition of uricase, as evidenced by plasma sUA (*Figure 3—figure supplement 3C*). Although short-term administration of OA alone failed to increase plasma sUA levels, we used it as the background for metabolic studies in mice to exclude potential interference.

One- or 3-day moderate sUA supplementation slightly but significantly increased whole blood NAD$^+$ levels in wild-type (WT) mice but not in CD38 knockout (KO) mice (*Figure 3A and B*; *Figure 3—figure supplement 5C and D*). Similar results were also observed under inflammatory conditions (*Figure 3—figure supplement 4A and B*; *Figure 4—figure supplement 4D and E*), and OA or inosine alone had no interference (*Figure 3—figure supplement 4E–H*). This indicates that CD38 mediates, at least in part, the suppressive effect of sUA on NAD$^+$ degradation. The suppression of NAD$^+$ degradation by the sUA-CD38 interaction was validated by a decrease in cADPR production under inflammatory conditions (*Figure 3—figure supplement 4D*) but not under non-inflammatory conditions (*Figure 3D*; *Figure 3—figure supplement 5F*), possibly because of physiological compensation via other cADPR synthases. However, sUA had a minor effect on NMN levels in vivo (*Figure 3C*; *Figure 3—figure supplement 4C*; *Figure 3—figure supplement 5E*; *Figure 4—figure supplement 4F*), likely due to the rapid conversion of NMN to NAD$^+$ (*Mills et al., 2016*). Notably, WT and CD38 KO mice showed similar NAD$^+$ baselines in whole blood, suggesting that the metabolic background has been reprogrammed in CD38 KO mice; for instance, sirtuins with higher activity in CD38 KO mice (*Camacho-Pereira et al., 2016*) may consume more NAD$^+$. Moreover, sUA release in vivo showed that both plasma sUA and whole blood NAD$^+$ gradually returned to their initial levels (*Figure 3E*), which confirmed the reversible regulation of NAD$^+$ availability by sUA and excluded potential interference from the gene regulation involved in other bioconversion pathways of NAD$^+$. Metabolic assays using recombinant hCD38 further verified that increased NAD$^+$ availability is mediated by direct CD38-sUA interaction (*Figure 3F*).

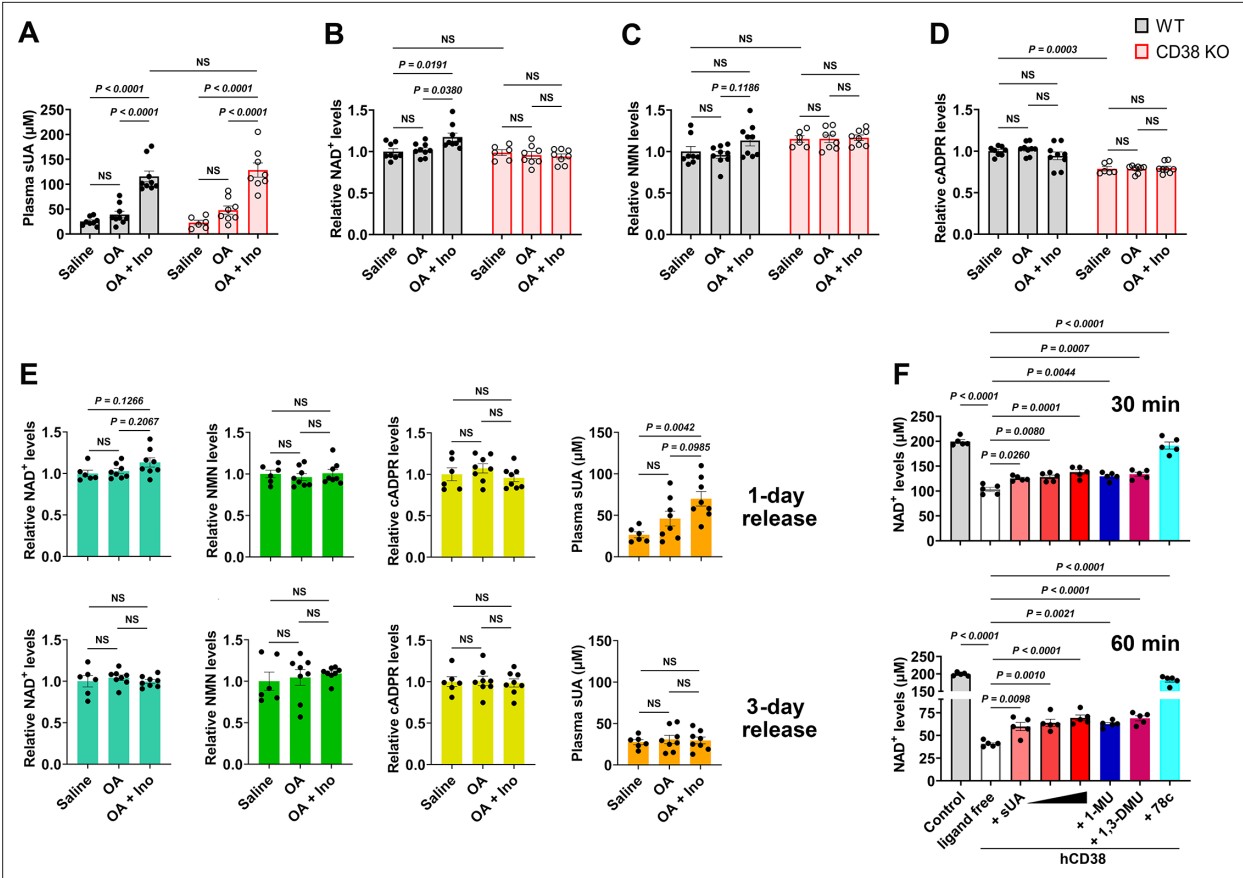

**Figure 3.** Soluble uric acid (sUA) physiologically limits NAD⁺ degradation via CD38 inhibition. Wild-type (WT) and CD38 knockout (KO) mice (10- to 12-week-old) received oral administration of saline, oxonic acid (OA), or OA plus inosine (Ino) twice (1-day moderate sUA supplementation). (**A–D**) Effect of 1-day sUA supplementation on plasma sUA (**A**) and whole blood NAD⁺ (**B**), nicotinamide mononucleotide (NMN) (**C**), and cyclic ADP-ribose (cADPR) (**D**) levels in WT and CD38 KO mice (WT-Saline: n=8 mice, WT-OA: n=9 mice, WT-OA+Ino: n=9 mice, KO-Saline: n=6 mice, KO-OA: n=8 mice, KO-OA+Ino: n=8 mice). (**E**) Effect of 1-day or 3-day release on whole blood NAD⁺, NMN, cADPR, and plasma sUA levels in WT mice that received 1-day supplementation (WT-Saline: n=6 mice, WT-OA: n=8 mice, WT-OA+Ino: n=8 mice). (**F**) Effect of sUA (100, 200, or 500 µM) and other ligands (analogs at 500 µM, 78c, a CD38 inhibitor, at 0.5 µM) on NAD⁺ degradation by recombinant hCD38 (n=5 independent samples). Data are mean ± s.e.m. Significance was tested using two-way ANOVA (**A–D**), Kruskal-Wallis test, or one-way ANOVA with Tukey's multiple comparisons test (**E and F**). NS, not significant. Statistical difference (**A–D**) between OA and OA+Ino groups in WT or KO mice (saline alone group excluded) was also analyzed by two-tailed unpaired t-test or Mann-Whitney test; WT mice: p<0.0001 (**A**), p=0.0056 (**B**), or 0.0351 (**C**); KO mice: p=0.0003 (**A**).

The online version of this article includes the following figure supplement(s) for figure 3:

**Figure supplement 1.** Effect of soluble uric acid (sUA) on intracellular NAD⁺ levels in A549 and THP-1 cells.

**Figure supplement 2.** Effect of soluble uric acid (sUA) and its precursors and metabolite on nicotinamide phosphoribosyltransferase (NAMPT) and poly(ADP-ribose) polymerase (PARP) activities.

**Figure supplement 3.** Effect of oxonic acid (OA) on CD38 activity and plasma soluble uric acid (sUA) levels.

**Figure supplement 4.** Effect of soluble uric acid (sUA) at physiological levels on NAD⁺ degradation under inflammatory conditions.

**Figure supplement 5.** Effect of 1-, 3-, and 7-day soluble uric acid (sUA) supplementation on NAD⁺, nicotinamide mononucleotide (NMN), and sUA levels in whole blood and tissues.

**Figure supplement 6.** No effect of 1- to 7-day soluble uric acid (sUA) supplementation on serum IL-1β production.

**Figure supplement 7.** Soluble uric acid (sUA) limits nicotinamide mononucleotide (NMN) degradation via CD38 inhibition.

In contrast, the tissue levels of NAD⁺ and NMN were not increased in 1- and 3-day sUA-supplementation models, probably because the tissue uptake of sUA was physiologically saturated before treatment (*Figure 3—figure supplement 5A and B*). The NAD⁺ levels in the brain and heart appeared to be elevated in 7-day model (*Figure 3—figure supplement 5G–I*). Thus, the sUA-CD38 interaction in the circulation may indirectly increase tissue NAD⁺ availability. Our models within 7 days

did not show an increase in serum IL-1β production (*Figure 3—figure supplement 6*), but we did not prolong the treatment time because long-term administration of OA might induce renal damage with concomitant inflammation related to cell death.

Considering the rapid conversion of NMN to NAD$^+$ in vivo, we assessed the effect of sUA on extracellular NMN degradation in vitro. sUA directly inhibited NMN degradation by recombinant hCD38 (*Figure 3—figure supplement 7A*). Whereas sUA or another CD38 inhibitor failed to boost intracellular NAD$^+$ in primed WT bone marrow-derived macrophages (BMDMs) treated with NMN (*Figure 3—figure supplement 7B*). This is likely because another NAD$^+$-consuming enzyme, PARP1, is activated (*Minhas et al., 2019*) and the NMN transporter expression is very low in BMDMs (*Chini et al., 2020*). Importantly, sUA increased extracellular NMN availability in WT BMDMs but not in CD38 KO BMDMs (*Figure 3—figure supplement 7C and D*). The addition of recombinant hCD38 restored the suppressive effect of sUA on extracellular NMN degradation in CD38 KO BMDMs (*Figure 3—figure supplement 7E*). Accordingly, sUA may also increase NAD$^+$ availability by inhibiting CD38-medeiated NMN degradation.

## sUA physiologically prevents excessive inflammation by interacting with CD38

NAD$^+$ is crucial for the activity of sirtuins that limit the NLRP3 inflammasome activation (*He et al., 2020*; *Misawa et al., 2013*), and cADPR may regulate calcium signaling to promote cytokine production (*Murakami et al., 2012*; *Zeidler et al., 2022*). Therefore, CD38 plays a key role in inflammation via NAD$^+$ metabolism (*Piedra-Quintero et al., 2020*). We confirmed the role of CD38 in the NLRP3 inflammasome activation. CD38 KO reduced IL-1β release driven by several inflammasome activators and the fungal component zymosan in primed BMDMs (*Figure 4—figure supplement 1A*). However, pre-incubation with sUA at physiological levels hardly suppressed inflammasome activation in primed WT BMDMs (*Figure 4—figure supplement 1B*). sUA uptake was very low in primed WT BMDMs (*Figure 4—figure supplement 1D*), suggesting a crucial role of intracellular sUA in regulating inflammasome activation in vitro. However, extracellular sUA may inhibit CD38-mediated NMN degradation to increase NAD$^+$ levels in vivo, thus limiting excessive inflammation via sirtuins signaling. In fact, we noticed that sUA pre-incubation moderately limited IL-1β release in primed THP-1 cells (*Figure 4—figure supplement 2A*), and CD38 blockade abrogated this effect without reducing sUA uptake (*Figure 4—figure supplement 2B and C*), which suggests CD38 as a key mediator in the immunosuppressive effect of sUA at physiological levels. Notably, sUA did not induce IL-1β release in primed WT BMDMs and THP-1 cells (*Figure 4—figure supplement 1C*; *Figure 4—figure supplement 2A*). The pro-inflammatory potential of sUA, especially at supraphysiological levels, has been challenged by the improper preparation of aqueous solution (MSU crystal precipitation) in basic studies (*Alberts et al., 2019*; *Ma et al., 2020*). We also confirmed that long-term storage at 4°C promoted crystal precipitation in high-concentration sUA stock solutions (*Figure 4—figure supplement 3*).

To reveal the role of sUA-CD38 interaction in regulating inflammation and innate immunity, at first, we stimulated WT and CD38 KO mice with crude lipopolysaccharide (cLPS) after 1-day sUA supplementation. OA was used as the background in mice to exclude its interference. Plasma sUA at the minimum physiological levels of humans (OA plus inosine group) suppressed cLPS-induced production of serum IL-1β and IL-18 in WT mice without affecting inflammasome-independent TNF-α levels, and the suppressive effects were abrogated in CD38 KO mice (*Figure 4A–C*), demonstrating that sUA physiologically limits cLPS-induced systemic inflammation via CD38. sUA immunosuppression may be partially mediated by increased NAD$^+$ levels and decreased cADPR production in whole blood after sUA-CD38 interaction (*Figure 3—figure supplement 4B and D*). sUA hardly limited high-dose cLPS-induced systemic inflammation (*Figure 4—figure supplement 4A–C*), although plasma sUA and whole blood NAD$^+$ levels were increased (*Figure 4—figure supplement 4D and E*). In addition, high-dose cLPS, with or without sUA supplementation, did not affect cADPR levels (*Figure 4—figure supplement 4G*), suggesting some CD38-independent inflammatory mechanisms.

Recently, crystal-free hyperuricemia has been shown to rapidly inhibit neutrophil migration (*Ma et al., 2022*), suggesting that some biological targets directly mediate sUA immunosuppression (*Lowell, 2022*). CD38 is crucial for neutrophil recruitment (*Partida-Sánchez et al., 2001*), we therefore investigated the effect of sUA on MSU crystal-induced peritonitis via CD38. After 1-day sUA supplementation, plasma sUA at the minimum physiological levels of humans (OA plus inosine group)

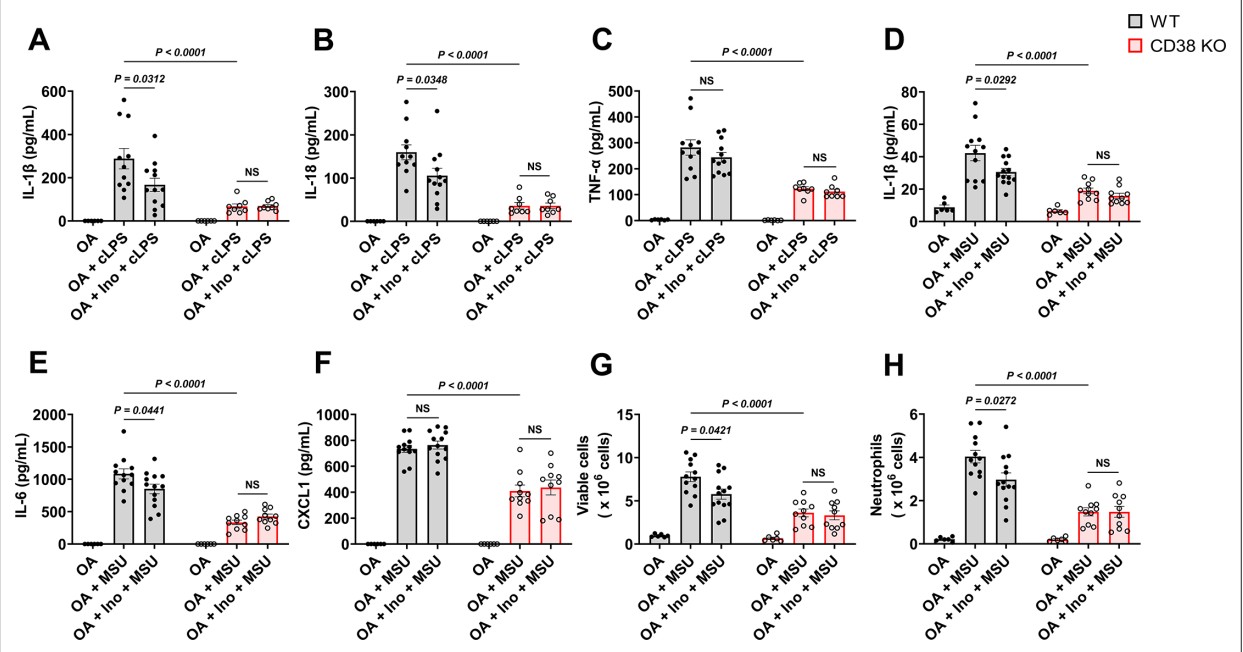

**Figure 4.** Soluble uric acid (sUA) physiologically prevents excessive inflammation by interacting with CD38. Wild-type (WT) and CD38 knockout (KO) mice (10- to 12-week-old) received 1-day moderate sUA supplementation, plasma sUA was increased to the minimum physiological levels of humans in oxonic acid (OA) plus inosine (Ino) groups. Then, the mice were stimulated with crude lipopolysaccharide (cLPS) (2 mg/kg) or monosodium urate (MSU) crystals (2 mg/mouse) for 6 hr. (A–C) Effect of sUA at physiological levels on serum levels of IL-1β (A), IL-18 (B), and TNF-α (C) in mice with cLPS-induced systemic inflammation (WT-OA: n=6 mice, WT-OA+cLPS: n=11 mice, WT-OA+Ino+cLPS: n=12 mice, KO-OA: n=6 mice, KO-OA+cLPS: n=8 mice, KO-OA+Ino+cLPS: n=8 mice). (D–H) Effect of sUA at physiological levels on IL-1β (D), IL-6 (E), and CXCL1 (F) levels and recruitment of viable cells (red blood cells excluded) (G) and neutrophils (H) in peritoneal lavage fluid from the mice with MSU crystal-induced peritonitis (WT-OA: n=6 mice, WT-OA+MSU: n=12 mice, WT-OA+Ino+MSU: n=13 mice, KO-OA: n=6 mice, KO-OA+MSU: n=10 mice, KO-OA+Ino+MSU: n=10 mice). Data are mean ± s.e.m. Significance was tested using two-way ANOVA with Tukey's multiple comparisons test. NS, not significant. Statistical difference between OA+cLPS (or MSU) and OA+Ino+cLPS (or MSU) groups in WT or KO mice (OA alone group excluded) was also analyzed by two-tailed unpaired t-test (with Welch's correction when applicable) or Mann-Whitney test; WT mice: p=0.0398 (A), 0.0383 (B), 0.0407 (D), 0.0417 (E), 0.0222 (G), or 0.0205 (H).

The online version of this article includes the following figure supplement(s) for figure 4:

**Figure supplement 1.** Effect of CD38 knockout (KO) or soluble uric acid (sUA) pre-incubation on IL-1β release in primed bone marrow-derived macrophages (BMDMs).

**Figure supplement 2.** Effect of soluble uric acid (sUA) pre-incubation and CD38 blockade on IL-1β release in primed THP-1 cells.

**Figure supplement 3.** Crystal precipitation in high-concentration soluble uric acid (sUA) stock solutions after long-term storage.

**Figure supplement 4.** One-day soluble uric acid (sUA) supplementation fails to prevent high-dose crude lipopolysaccharide (cLPS)-induced systemic inflammation.

**Figure supplement 5.** Oxonic acid (OA) or inosine (Ino) alone does not limit crude lipopolysaccharide (cLPS)-induced systemic inflammation and monosodium urate (MSU) crystal-induced peritonitis.

**Figure supplement 6.** Potential mechanism of the paradox in gout therapy (urate-lowering therapy).

inhibited the recruitment of viable cells and neutrophils and reduced the production of IL-1β and IL-6 in WT mice but not in CD38 KO mice (*Figure 4D, E, G, and H*), which verified the involvement of CD38 in the suppressive effect of sUA on inflammation and innate immunity. CXCL1 production (*Figure 4F*) was not affected by sUA, indicating that the sUA-CD38 interaction may inhibit circulating immune cells in response to chemokines. Moreover, sUA may interfere with the interaction between CD38 and adhesion molecules such as CD31 (*Deaglio et al., 2010*; *Deaglio et al., 1998*) to suppress immune cell migration. Intriguingly, intact IL-8 signaling has been reported in CD38 KO mice (*Partida-Sánchez et al., 2001*), and sUA at a high concentration (595 μM) inhibits neutrophils in response to IL-8 (*Ma et al., 2022*). Whereas we did not observe decreased neutrophil recruitment by sUA in CD38 KO mice, likely because plasma sUA levels (around 120 μM) in our models are insufficient to affect IL-8 signaling. In addition, we cannot exclude the possibility that sUA at supraphysiological levels interacts

with other targets to regulate IL-8 signaling and β2-integrin, which requires further investigation in the future.

Inosine, an sUA precursor, may also exhibit anti-inflammatory effect by interacting with adenosine receptors (*Gomez and Sitkovsky, 2003*). Scott et al. reported that after oral administration, serum inosine concentrations slightly increased and then rapidly returned to the initial levels in mice within 2 hr (*Scott et al., 2002*). We stimulated the mice with cLPS or MSU crystals 2 hr after the second treatment, suggesting the negligible contribution of inosine. To verify this and exclude the potential contribution of OA, we evaluated the effects of 1-day treatment of inosine or OA on inflammation in vivo. The results showed that OA or inosine alone hardly affected cLPS-induced systemic inflammation (*Figure 4—figure supplement 5A–C*) and MSU crystal-induced peritonitis (*Figure 4—figure supplement 5D–H*), as well as plasma sUA levels and whole blood $NAD^+$ metabolism under inflammatory conditions (*Figure 3—figure supplement 4E–H*). Thus, these results suggested that sUA at physiological levels limits innate immunity to avoid excessive inflammation by interacting with CD38.

## Discussion

In the present study, we unveiled CD38 as a direct physiological target for sUA and thus defined its fundamental physiological functions in the regulation of $NAD^+$ availability and innate immunity, which promotes understanding of the molecular basis of sUA physiology as well as providing an important clue to explore the potential impact of abnormal sUA levels (independent of MSU crystals) on health and disease.

It has been reported that $NAD^+$ decline contributes to inflammation, aging-related dysfunction, and multiple diseases, including hearing loss, obesity, diabetes, kidney diseases, and cardiovascular diseases, in murine models (*Hogan et al., 2019*) and possibly even in humans (*Rajman et al., 2018*). Accordingly, $NAD^+$ boosting by CD38 inhibitors has been a promising therapeutic strategy (*Chini et al., 2020*; *Chini et al., 2018*; *Escande et al., 2013*; *Hogan et al., 2019*; *Peclat et al., 2020*; *Tarragó et al., 2018*). We discovered a structural feature for pharmacological inhibition of CD38 based on sUA analogs such as caffeine metabolites. Indeed, sUA and its analogs have similar functions, such as antioxidant property (*Nishida, 1991*) and neuroprotective effects (*Haberman et al., 2007*), supporting that they share a functional group, 1,3-DHI-2-one, that interacts with the same targets, including but not limited to CD38. Importantly, our results support that sUA at physiological levels limits CD38 activity to maintain $NAD^+$ availability, providing the molecular basis for sUA preventing $NAD^+$ decline-associated senescence and diseases. sUA levels seem to increase with age (*Iwama et al., 2012*; *Kuzuya et al., 2002*), raising a possibility that sUA elevation within physiological range is a compensatory response to aging in organisms. Given that CD38 expression is relatively low in young mice (*Camacho-Pereira et al., 2016*), aged mice would be more appropriate to test this hypothesis. The apparent $K_i$ values imply that CD38 is completely inhibited by sUA in blood or in some tissues under physiological conditions; however, this might not be the case because other endogenous substance may also regulate CD38 activity (*Chini et al., 1997*; *Dogan et al., 2002*; *Hara-Yokoyama et al., 1996*). In spite of this, altered sUA levels are able to indicate the changes in the activities of CD38 and other molecular targets of sUA, which partially explains the correlation between sUA homeostasis disruption and disease risk. For instance, abnormal reduction of sUA levels such as hypouricemia may result in higher CD38 activity in the circulation due to the reversible inhibitory effect of sUA, thus negatively influencing health as well as increasing the risk of certain diseases. However, it does not mean that higher sUA levels are better, because excessive elevation of sUA levels promotes the precipitation of MSU crystals. In addition, we cannot exclude the possibility that long-term and crystal-free hyperuricemia (>420 µM) in humans may overly modulate additional unknown targets, especially in CD38-negative cells, thereby partially covering the CD38-mediated physiological functions of sUA. To identify more molecular targets for sUA, drug affinity responsive target stability (*Lomenick et al., 2009*) and cellular thermal shift assay (*Martinez Molina et al., 2013*), which detect the direct interactions between binding proteins and their ligands, may serve as alternative strategies.

Another unexpected observation is the restrictive effect of sUA at physiological levels on excessive inflammation, which is complementary to several recent studies regarding the immunosuppressive effect of crystal-free hyperuricemia in the host (*Ma et al., 2020*; *Ma et al., 2022*). Similar to itaconate, an inhibitor of the NLRP3 inflammasome (*Hooftman et al., 2020*), sUA production increases in activated macrophages (*Ives et al., 2015*). It has been reported that intracellular sUA reduction promotes

certain inflammatory responses (*Ives et al., 2015*; *Kono et al., 2010*), hence sUA at physiological levels may function as an endogenous regulator of inflammasomes to avoid excessive inflammation by inhibiting CD38 activity. On the other hand, chemical phase transition (crystal precipitation) is essential for sUA as a danger signal to trigger immune responses (*Alberts et al., 2019*; *Ives et al., 2015*; *Ma et al., 2020*; *Ma et al., 2022*; *Shi, 2010*; *Shi et al., 2003*). We and another laboratory demonstrated that MSU crystals upregulate CD38 to promote inflammatory responses in primed macrophages (*Wen et al., 2021*; *Yan, 2021*). A recent study confirmed CD38 upregulation in gout patients and further validated the role of CD38 in experimental models of gouty inflammation induced by MSU crystals (*Alabarse et al., 2024*). Importantly, in this study, we showed that CD38 mediates the opposite effects of sUA (especially at physiological levels) and MSU crystals on inflammation and innate immunity. Therefore, a sudden and rapid reduction of the circulating sUA levels by high-dose urate-lowering medications may disrupt the immune balance by rapidly releasing CD38 activity before MSU crystal dissolution (*Figure 4—figure supplement 6*), resulting in the increased risk of gout attack in the initiation of therapy (*Becker et al., 2005*; *Wen et al., 2024*), a well-known paradox in gout therapy. Several clinical studies even observed that blood sUA levels decrease during gout flares, and the resolution of gouty inflammation is coming with a recovery of sUA levels (*Logan et al., 1997*; *Urano et al., 2002*). Given the distinct effects of sUA and MSU crystals on certain targets, intracellular and/or extracellular microcrystals should be excluded when evaluating the pathological role of sUA at high levels in relevant studies. In addition, our findings also provide biological evidence for the neuroprotection of sUA. It has been reported that CD38 is involved in neurodegenerative diseases (*Blacher et al., 2015*; *Meyer et al., 2022*); thus, sUA in central nervous system tissues may directly inhibit CD38 activity to limit neuroinflammation and the progression of such diseases.

Previously, a physiological medium containing sUA was shown to inhibit UMP synthase to reshape cellular metabolism in vitro (*Cantor et al., 2017*). We identified a unique sUA-CD38 interaction in this study, highlighting the physiologically essential role of sUA as a purine metabolite in sustaining life. Accumulated data suggests a remarkable overlap between the effects of sUA (*Álvarez-Lario and Macarrón-Vicente, 2010*; *Ames et al., 1981*; *Cutler et al., 2019*; *Kutzing and Firestein, 2008*; *Lai et al., 2017*; *Linnerz et al., 2022*; *Lu et al., 2016*; *Ma et al., 2020*; *Ma et al., 2022*; *Scott et al., 2002*; *Wan et al., 2020*) and CD38 inhibition/KO (*Blacher et al., 2015*; *Chini et al., 2018*; *Hogan et al., 2019*; *Meyer et al., 2022*) in counteracting excessive inflammation, aging, and certain diseases, which also strongly supports our current findings. It should be noticed that we used an exogenous compound as the background in mice to mimic the deficiency of uricase in humans, and tissue sUA levels were unchanged after sUA supplementation, suggesting some of the limitations in our models when evaluating the physiological relevance. Therefore, it is important to extend these studies to global sUA-depletion models in primates or uricase-transgenic mice. Moreover, the commercial recombinant hCD38 without a transmembrane region was partially used in this study, we cannot exclude the potential interaction between sUA and the transmembrane region of CD38, although comparable $K_i$ values were obtained in crude enzymes containing full-length CD38. In addition to identification of the allosteric sites of CD38, the crystallographic structure of the active full-length CD38-sUA complex should be captured using cryo-electron microscopy to elucidate the inhibitory mechanisms in the future. However, the present study clearly demonstrated that sUA at physiological levels directly inhibits CD38 and consequently limits $NAD^+$ degradation and excessive inflammation, suggesting that sUA is crucial for the physiological defense in humans against aging and diseases.

## Materials and methods

### Key resources table

| Reagent type (species) or resource | Designation | Source or reference | Identifiers | Additional information |
|---|---|---|---|---|
| Chemical compound, drug | Uric acid | Sigma-Aldrich | Cat#U0881 | |
| Chemical compound, drug | β-Nicotinamide Adenine Dinucleotide (NAD+) | Nacalai | Cat#24338-86 | |

*Continued on next page*

*Continued*

| Reagent type (species) or resource | Designation | Source or reference | Identifiers | Additional information |
|---|---|---|---|---|
| Chemical compound, drug | β-Nicotinamide mononucleotide (NMN) | BLD Pharmatech | Cat#BD116593 | |
| Chemical compound, drug | cADP-Ribose (cADPR) | Santa Cruz | Cat#sc-201512 | |
| Chemical compound, drug | FYU-981 | Fuji Yakuhin Co., Ltd. | N/A | |
| Chemical compound, drug | 78c | Sigma-Aldrich | Cat#5.38763 | |
| Chemical compound, drug | *N*-Cyclohexyl benzamide | Sigma-Aldrich | Cat#R531332 | |
| Chemical compound, drug | Inosine | Nacalai | Cat#07139-42 | |
| Chemical compound, drug | Hypoxanthine | Sigma-Aldrich | Cat#H-9636 | |
| Chemical compound, drug | Xanthine | Sigma-Aldrich | Cat#X0626 | |
| Chemical compound, drug | Allantoin | Sigma-Aldrich | Cat#A-7878 | |
| Chemical compound, drug | Adenosine | Wako | Cat#010-10491 | |
| Chemical compound, drug | Guanosine | Wako | Cat#079-01111 | |
| Chemical compound, drug | Uracil | FUJIFILM Wako | Cat#212-00062 | |
| Chemical compound, drug | 1,3-Dihydroimidazol-2-one | BLD Pharmatech | Cat#BD00733674 | |
| Chemical compound, drug | Oxypurinol | Sigma-Aldrich | Cat#O-6881 | |
| Chemical compound, drug | Caffeine | FUJIFILM Wako | Cat#031-06792 | |
| Chemical compound, drug | 1-Methyluric acid | Sigma-Aldrich | Cat#M6885 | |
| Chemical compound, drug | 1,3-Dimethyluric acid | Santa Cruz | Cat#sc-206240 | |
| Chemical compound, drug | 1,7-Dimethyluric acid | Cayman Chemical | Cat#19584 | |
| Chemical compound, drug | 1,3,7-Trimethyluric acid | Cayman Chemical | Cat#16949 | |
| Chemical compound, drug | 8-Oxoguanine | Cayman Chemical | Cat#89290 | |
| Chemical compound, drug | Oxonic acid potassium salt | Sigma-Aldrich | Cat#156124 | |
| Chemical compound, drug | LPS from *Escherichia coli* 0111:B4 (crude LPS) | Sigma-Aldrich | Cat#L4130 | |
| Chemical compound, drug | Ultrapure LPS from *Escherichia coli* 0111:B4 | InvivoGen | Cat#tlrl-3pelps | |
| Chemical compound, drug | Phorbol 12-Myristate 13-Acetate | FUJIFILM Wako | Cat#162-23591 | |

*Continued on next page*

*Continued*

| Reagent type (species) or resource | Designation | Source or reference | Identifiers | Additional information |
|---|---|---|---|---|
| Chemical compound, drug | Nigericin | Sigma-Aldrich | Cat#N-7143 | |
| Chemical compound, drug | ATP | Oriental Yeast | Cat#45142000 | |
| Chemical compound, drug | Zymosan A from *Saccharomyces cerevisiae* | Sigma-Aldrich | Cat#Z4250 | |
| Chemical compound, drug | Nicotinamide 1, $N^6$-ethenoadenine dinucleotide (ε-NAD$^+$) | Sigma-Aldrich | Cat#N2630 | |
| Chemical compound, drug | Nicotinamide guanine dinucleotide (NGD) | Sigma-Aldrich | Cat#N5131 | |
| Peptide, recombinant protein | Recombinant human CD38 protein | Sino Biological | Cat#10818-H08H | |
| Peptide, recombinant protein | Recombinant human M-CSF protein | Proteintech Group | Cat#HZ-1192 | |
| Commercial assay or kit | Human IL-1β ELISA Kit | R&D Systems | Cat#DY201 | |
| Commercial assay or kit | Mouse IL-1β ELISA Kit | R&D Systems | Cat#SMLB00C | |
| Commercial assay or kit | Mouse IL-6 ELISA Kit | R&D Systems | Cat#DY406-05 | |
| Commercial assay or kit | Mouse IL-18 ELISA Kit | R&D Systems | Cat#DY7625-05 | |
| Commercial assay or kit | Mouse TNF-α ELISA Kit | R&D Systems | Cat#DY410-05 | |
| Commercial assay or kit | Mouse CXCL1 ELISA Kit | R&D Systems | Cat#DY453-05 | |
| Commercial assay or kit | Wright-Giemsa Stain Kit | ScyTek Laboratories | Cat#WGK-1 | |
| Commercial assay or kit | CycLex NAMPT Colorimetric Assay Kit Ver.2 | MBL | Cat#CY-1251V2 | |
| Commercial assay or kit | HT Universal Colorimetric PARP Assay Kit | R&D Systems | Cat#4677-096-K | |
| Cell line (human) | A549 | ATCC | CCL-185 | |
| Cell line (human) | THP-1 | ATCC | TIB-202 | |
| Cell line (mouse) | ICR bone marrow-derived macrophages | This paper | N/A | See more details in Preparation of BMDMs section |
| Strain, strain background (mouse) | ICR mice | Japan SLC, Inc | N/A | |
| Strain, strain background (mouse) | CD38 KO ICR mice | Dr. Haruhiro Higashida, Kanazawa University | N/A | |
| Software, algorithm | Prism 9 | GraphPad Software | https://www.graphpad.com | |
| Software, algorithm | Labsolutions software | Shimadzu | https://www.shimadzu.com | |
| Other | RF-6000 Spectrofluorophotometer | Shimadzu | https://www.shimadzu.com | For the measurement of CD38 activity |
| Other | LC-30A system | Shimadzu | https://www.shimadzu.com | For LC-MS/MS analysis |
| Other | LCMS-8050 | Shimadzu | https://www.shimadzu.com | For LC-MS/MS analysis |

## Animals

Male and female ICR (Institute of Cancer Research of the Charles River Laboratories, Inc, Wilmington, MA, USA) mice were initially purchased from Japan SLC, Inc. CD38 KO mice (ICR strain) were generated using the CRISPR/Cas9 method as previously described (*Ichinose et al., 2019*). WT and CD38

KO mice were kept and bred at the Experimental Animal Center of Kanazawa University (Takara-machi campus). For animal experiments, all the mice (sex and age as indicated in respective figure legends) were transferred a week in advance and housed in the animal room of Research Center of Child Mental Development under standard conditions (24°C; 12 hr light/dark cycle, lights on at 8:30 am) with standard chow and water provided ad libitum. Male and female mice were separated after weaning and were equally used for experiments except when specified. In each experimental group, the mice were from different biological mothers. All animal experiments were approved by the Institutional Animal Care and Use Committee at Kanazawa University (AP-214243), and were performed in accordance with ARRIVE and the local guidelines.

## Cell culture

THP-1 and A549 cells were cultured in RPMI-1640 containing 10% FBS and 1% penicillin/streptomycin. Bone marrow cells and BMDMs were maintained in RPMI-1640 containing 10% FBS, 1% penicillin/streptomycin, and 50 μM 2-mercaptoethanol; macrophage colony stimulating factor (M-CSF) was used in the preparation of BMDMs.

## Measurement of CD38 activity

The hydrolase activity of CD38 was measured according to a previous report (*de Oliveira et al., 2018*) with minor modifications. CD38 hydrolase activity was measured using 50 μM ε-NAD$^+$ as a substrate in hydrolase reaction buffer (250 mM sucrose, 40 mM Tris-HCl, pH 7.4). Briefly, cells or tissues were directly homogenized in blank reaction buffer on ice, and recombinant hCD38 was diluted in blank reaction buffer for subsequent assays. The tissue homogenates were centrifuged to collect the supernatant for enzyme assays. The loading volume of enzyme was 4–30 μL, the total volume of the reaction system was 3 mL. To detect enzyme inhibition, except when specified, the ligands were directly dissolved in hydrolase reaction buffer before pH adjustment, after which, the pH was immediately adjusted to 7.4. All reaction buffers, with or without ligands, were freshly prepared before each assay. For measurement of hydrolase activity, 3 mL reaction buffer containing enzyme, ε-NAD$^+$, and ligands at indicated concentrations was maintained at 37°C with constant stirring.

The cyclase activity of CD38 was measured as previously described (*Higashida et al., 1999*; *Jin et al., 2007*). CD38 cyclase activity was measured using 60 μM NGD as a substrate in cyclase reaction buffer (100 mM KCl, 10 μM CaCl$_2$, and 50 mM Tris-HCl, pH 6.6). Tissues were cut into pieces and suspended in the buffer (5 mM MgCl$_2$, 10 mM Tris-HCl, pH 7.3) at 4°C for 30 min. Then, the suspension was homogenized on ice, and the supernatant was collected after centrifugation. To collect the crude membrane fractions, the supernatant was centrifuged at 105,000×$g$ for 30 min. The final pellet was resuspended in 10 mM Tris-HCl solution (pH 6.6) for cyclase assay. Homogenates of THP-1 or A549 cells, and recombinant hCD38 were used directly for cyclase assays. The loading volume of enzyme was 4–20 μL, the total volume of reaction system was 3 mL. To detect enzyme inhibition, except when specified, the ligands were directly dissolved in cyclase reaction buffer before pH adjustment, after which, the pH was immediately adjusted to 6.6. All reaction buffers, with or without ligands, were freshly prepared before each assay. For measurement of cyclase activity, 3 mL reaction buffer containing enzyme, NGD, and ligands at indicated concentrations was maintained at 37°C with constant stirring.

The reaction buffer for hydrolase or cyclase assays was excited at 300 nm, and fluorescence emission was measured every second at 410 nm by Shimazu RF-6000 spectrofluorometer. Hydrolase or cyclase activity was calculated from the linear portion of the time course by fitting a linear function to the data points recorded within 5 min.

In this study, 8-OG and guanosine were tested only at 50 μM due to the limited solubility. Guanosine was dissolved in DMSO and diluted in the reaction buffer for subsequent assays. Other ligands such as sUA, inosine, hypoxanthine, xanthine, allantoin, adenosine, uracil, 1,3-DHI-2-one, oxypurinol, caffeine, 1-MU, 1,3-DMU, 1,7-DMU, and 1,3,7-TMU were tested at the concentrations as indicated.

For reversibility test, recombinant hCD38 was pre-incubated for 30 min in four conditions: (1) control reaction buffer; (2) in the presence of 100 μM substrate (ε-NAD$^+$ or NGD); (3) 500 μM sUA; (4) both substrate and sUA. Subsequently, the enzyme was diluted 100-fold in reaction buffer in the presence or absence of 500 μM sUA for activity assays using ε-NAD$^+$ or NGD. Naïve THP-1 and A549 cells were incubated with RPMI-1640 medium in the presence or absence of 500 μM sUA for 2 hr. The cells

were collected and homogenized in the reaction buffer on ice with or without 500 µM sUA, samples were then diluted 100-fold in reaction buffer with or without 500 µM sUA for enzyme assay. Control group was not treated with sUA in all steps; sUA group was treated with sUA in each step; sUA release group was treated with sUA prior to dilution in sUA-free buffer for enzyme assays.

## Measurement of NAMPT and PARP activity

The activities of NAMPT and PARP were measured using commercial kits, HT Universal Colorimetric PARP Assay Kit, and CyclLex NAMPT Colorimetric Assay Kit Ver.2. The solutions of ligands were freshly prepared (pH was adjusted to 7.4) and were immediately used for enzyme assays.

## Moderate sUA supplementation in mice

Plasma sUA levels in mice were increased to the minimum physiological levels of humans by moderate sUA supplementation. For 1-day sUA supplementation, WT and CD38 KO mice received oral administration of saline, OA (1.5 g/kg), or OA (1.5 g/kg) plus inosine (1.5 g/kg), the gavage volume was 5 mL/kg. Drug suspension in saline was freshly prepared and warmed to 37°C before each treatment. The mice received the first treatment on the evening (19:00) of the first day and the second treatment on the morning (9:00) of the second day. For 3- or 7-day sUA supplementation, WT and CD38 KO mice received the same treatment twice daily from the evening of the first day to the morning of the last day. Four hours after the last treatment, the mice were sacrificed and whole blood, serum, plasma, and tissues were collected for metabolic studies. For immunological studies, mice were stimulated with different ligands 2 hr after the second treatment on the morning (9:00) of the second day.

## sUA release in mice

WT mice received oral administration of saline, OA, or OA plus inosine (1-day supplementation model, as described above). One day (28 hr) or 3 days (76 hr) after the second treatment, the mice were sacrificed and whole blood and plasma were collected.

## cLPS-induced systemic inflammation

Plasma sUA levels in WT and CD38 KO mice were increased to the minimum physiological levels of humans by 1-day sUA supplementation (OA plus inosine). Two hours after the last treatment of OA, or OA plus inosine, the mice were intraperitoneally injected with sterile PBS or cLPS (2 or 20 mg/kg), the injection volume was 3 mL/kg. Four hours (20 mg/kg) or 6 hr (2 mg/kg) after stimulation, the mice were sacrificed and whole blood, plasma, and serum were collected.

## MSU crystal-induced peritonitis

Plasma sUA levels in WT and CD38 KO mice were increased to the minimum physiological levels of humans by 1-day sUA supplementation (OA plus inosine). Two hours after the last treatment of OA, or OA plus inosine, the mice were intraperitoneally injected with sterile PBS or MSU crystals (2 mg/mouse), and the injection volume was 200 µL/mouse. Six hours after stimulation, the mice were sacrificed and blood samples were collected. For each mouse, the peritoneal cavity was washed with 5 mL ice-cold sterile PBS, the supernatant was collected by centrifugation for subsequent ELISA, and cell pellets were used for total viable cell count by Trypan Blue staining. In brief, cell pellet from 1.5 mL of peritoneal lavage fluid was resuspended in RBC lysis buffer for 30 s, sterile PBS (ninefold volume) was added to terminate lysis. Afterward, the cells were centrifuged at 1000 rpm for 5 min and resuspended in sterile PBS for viable cell count. For neutrophil count, peritoneal lavage fluid was directly used for smears after appropriate dilution, and subsequently, Wright-Giemsa staining was performed according to the protocol provided by the manufacturer. Viable cells and neutrophils were counted by two investigators (one investigator did not participate in this project and was blind to the information of experimental groups), and the mean numbers were shown in the figures.

## ELISA

Human IL-1β, mouse IL-1β, IL-6, IL-18, TNF-α, and CXCL1 levels were measured according to the protocols provided by the manufacturer. Serum samples were diluted before ELISA when applicable. In this study, all the samples were frozen after collection and thawed before ELISA.

## Preparation of MSU crystals and sUA solution

MSU crystals were prepared by the recrystallization of oversaturated sUA according to a previous report (*Martinon et al., 2006*). Improper preparation of sUA solution may introduce crystals to cause false-positive or false-negative results. It has been demonstrated that crystals may precipitate in sUA solution prepared by pre-warming to activate immune cells (*Ma et al., 2020*). In this study, the sUA solution was prepared according to an improved protocol (*Ma et al., 2020*). Briefly, we directly dissolved sUA at 0.5 mg/mL in blank RPMI-1640 medium by addition of NaOH, and adjusted the pH by HCl. Then, sUA solution was immediately filtered by 0.2 µm filters and diluted to experimental concentration (up to 10 mg/dL, 595 µM) for cell experiments. For all experiments, sUA solution was freshly prepared and used immediately.

## Preparation of BMDMs

Bone marrow cells were isolated from 10- to 12-week-old WT or CD38 KO mice (both male and female) by washing the marrow cavity with sterile PBS. The collected bone marrow cells were filtered by 70 µm strainers and then centrifuged at 1000 rpm, 4°C for 5 min. The cell pellet was resuspended in 2 mL red blood cell (RBC) lysis buffer for 30 s, then 8 mL complete RPMI-1640 medium (10% FBS, 1% penicillin/streptomycin, and 50 µM 2-mercaptoethanol) was added to terminate the lysis. After 5 min centrifugation at 1000 rpm, 4°C, the cells were resuspended in fresh complete RPMI-1640 medium and maintained for 4 hr in an incubator. Adherent cells were discarded, whereas non-adherent cells were cultured in complete RPMI-1640 medium containing 20 ng/mL M-CSF. After 3-day differentiation, the medium was replaced with fresh complete RPMI-1640 medium containing 20 ng/mL M-CSF. On the 7th day, BMDMs were collected for subsequent experiments. BMDMs were primed with 100 ng/mL ultrapure LPS for 4 hr for canonical inflammasome assay. For metabolic assay of NMN, BMDMs were primed with 100 ng/mL ultrapure LPS for 8 hr to induce higher protein expression of CD38.

## Intracellular NAD$^+$ assay

A549 cells were seeded in 24-well plates for NAD$^+$ assay. Briefly, when the confluence reached 80%, the culture medium in each well was discarded, and the cells were washed twice with sterile PBS and incubated in RPMI-1640 medium containing 1% FBS in the presence or absence of sUA (from 100 to 500 µM) for 20 hr. Then, the cells were washed twice with sterile PBS, after the second washing, PBS was completely removed and 100 µL 5% ice-cold perchloric acid (PCA) was added into each well. The plate was kept on ice for 2 hr, then cell samples were collected and centrifuged at 15,000 rpm, 4°C for 10 min. The supernatant was used for subsequent handling and measurement (see LC-MS/MS analysis).

Naïve THP-1 cells were pre-incubated with sUA (0–10 mg/dL) in RPMI-1640 medium containing 1% FBS for 2 hr, then the cells were washed twice with sterile PBS and stimulated with MSU crystals, cLPS, zymosan, or ATP for 6 hr. Subsequently, the cells were washed twice with sterile PBS, and then a total of 100 µL 5% ice-cold PCA was used to extract NAD$^+$ from cells in each well and medium as mentioned above.

WT BMDMs were primed with 100 ng/mL ultrapure LPS for 8 hr. Then, the cells were washed twice with sterile PBS. Afterward, the cells were incubated in control or 100 µM NMN-supplemented RPMI-1640 medium in the presence of sUA or 78c. After 6 hr incubation, the cells were washed twice with sterile PBS. Finally, cell samples for NAD$^+$ measurement were collected by 5% ice-cold PCA as mentioned above.

## Canonical inflammasome assay

Naïve THP-1 cells were primed with 0.5 µM phorbol 12-myristate 13-acetate for 3 hr the day before stimulation. Primed THP-1 cells were pre-incubated in RPMI-1640 medium in the presence or absence of sUA (5 or 10 mg/dL) for 2 hr. The cells were then washed twice with sterile PBS, and were stimulated with MSU crystals, cLPS, zymosan, and ATP in serum-free RPMI-1640 medium for 4 hr.

WT and CD38 KO BMDMs were primed with 100 ng/mL ultrapure LPS for 4 hr. Subsequently, primed BMDMs were pre-incubated with or without sUA for 2 hr, the cells were then washed twice with sterile PBS and were stimulated with nigericin, MSU crystals, or cLPS in serum-free RPMI-1640 medium.

After stimulation, the culture medium was collected and centrifuged at 3000 rpm, 4°C for 5 min, the supernatant was collected and stored at –30°C until ELISA.

## sUA uptake assay

WT BMDMs were primed with 100 ng/mL ultrapure LPS for 4 hr. Then, the cells were washed twice with sterile PBS, and maintained in RPMI-1640 medium containing sUA (100, 200, or 500 μM) for 2 or 15 hr. Subsequently, the cells were washed twice with ice-cold sterile PBS to terminate uptake, 100 μL 5% ice-cold PCA was added into each well. The plates were placed on ice for 2 hr, then cell samples were collected and centrifuged at 15,000 rpm, 4°C for 10 min. The supernatant was used for subsequent handling and measurement (see LC-MS/MS analysis).

## Metabolic assay of NAD$^+$

NAD$^+$ degradation by recombinant hCD38 was detected in hydrolase reaction buffer. At first, sUA (100, 200, and 500 μM) or other ligands at 500 μM in hydrolase reaction buffer (250 mM sucrose, 40 mM Tris) was freshly prepared, then the pH was immediately adjusted to 7.4 by HCl. Recombinant hCD38 was added into the buffer of experimental groups at 20 ng/mL, then the buffer was maintained at 37°C. The substrate NAD$^+$ was dissolved in hydrolase reaction buffer (pH 7.4) at 10 mM, and the pH was further adjusted to 7.4. The reaction was started with the addition of NAD$^+$ in the buffer for each group (final concentration is 200 μM), including control group (recombinant hCD38 free). All the buffers were incubated at 37°C with constant stirring. After 30 or 60 min, 20 μL reaction buffer was transferred into 180 μL of 5% ice-cold PCA, then vortexed for 30 s before 10 min centrifugation at 15,000 rpm, 4°C. The supernatant was further handled for NAD$^+$ measurement within 12 hr without freezing and thawing (see LC-MS/MS analysis).

## Metabolic assay of NMN

To prepare the reaction buffer, sUA was directly dissolved in blank RPMI-1640 medium by addition of NaOH, and the pH was immediately adjusted by HCl. After pH adjustment, the medium containing sUA was filtered by 0.2 μm filter and diluted as indicated. The medium containing 20 ng/mL recombinant hCD38 was then placed in a cell incubator. The reaction was started with the addition of NMN (final concentration was 200 μM). After incubation for 6 hr, 20 μL medium was transferred into 180 μL 5% ice-cold PCA, then vortexed for 30 s and centrifuged at 15,000 rpm, 4°C for 10 min. The supernatant was stored at –80°C until further handling for NMN measurement (see LC-MS/MS analysis).

## Extracellular NMN degradation

WT and CD38 KO BMDMs were primed with 100 ng/mL ultrapure LPS for 8 hr, then the cells were washed twice with sterile PBS. Primed BMDMs were maintained in RPMI-1640 medium supplemented with 100 μM NMN in the presence or absence of sUA. To restore the inhibitory effects of sUA on NMN degradation in KO BMDMs, recombinant hCD38 was added to the medium (final concentration was 10 ng/mL). After 6 hr incubation, culture medium was collected and centrifuged at 3000 rpm, 4°C for 5 min. Then, 20 μL of supernatant was transferred into 180 μL 5% ice-cold PCA, and was vortexed for 30 s and centrifuged at 15,000 rpm, 4°C for 10 min. The supernatant was stored at –80°C until further handling for NMN measurement.

## Handling of animal samples

The collected whole blood samples were immediately diluted 10-fold in 5% ice-cold PCA and homogenized on ice. After 10 min centrifugation at 15,000 rpm, 4°C, the supernatant was subpackaged for measurement or –80°C storage. For NAD$^+$ measurement in whole blood, the supernatant was handled without freezing and thawing and was measured within 24 hr of sample collection. For measurement of plasma sUA, after sample collection, we immediately diluted the plasma with 5% ice-cold PCA, after vortex, samples were centrifuged at 15,000 rpm, 4°C for 10 min, the collected supernatant was used for the subsequent handling and measurement within 24 hr.

After the collection of blood samples, the mice were immediately perfused with ice-cold sterile PBS. The tissue samples were collected, dried with tissue paper, and weighed. All the tissue samples were immediately homogenized in 5% ice-cold PCA on ice, then centrifuged at 15,000 rpm, 4°C for

10 min. The supernatant was collected and stored at –80°C until further handling for measurement (see LC-MS/MS analysis).

## LC-MS/MS analysis

As mentioned above, samples (cells, tissues, blood, or reaction buffer) were treated with ice-cold PCA, after extraction and centrifugation, the collected supernatant was appropriately diluted in 5% ice-cold PCA when applicable, and then was vortexed for 30 s and centrifuged at 15,000 rpm, 4°C for 5 min. Afterward, 30 μL supernatant was added into 200 μL 5 mM ammonium formate containing internal standards at indicated concentrations. Then, all the samples were vortexed for 30 s and centrifuged at 15,000 rpm, 4°C for 5 min again, the final supernatant was used for subsequent measurement. All the samples after handling were immediately analyzed in this study without freezing and thawing.

The LC-MS/MS system consisted of a triple quadrupole LCMS-8050 (Shimadzu) and an LC-30A system (Shimadzu). $NAD^+$, NMN, and the $N$-cyclohexyl benzamide (NCB, internal standard, 5 ng/mL) were eluted on Altlantis HILIC Silica (2.1×150 mm, 5 μm) at 40°C using an isocratic mobile phase containing 60% water with 0.1% formic acid and 40% acetonitrile with 0.1% formic acid at 0.4 mL/min. The selected transitions of m/z were $664.10 \rightarrow 136.10$ for $NAD^+$, $334.95 \rightarrow 123.15$ for NMN, and $204.10 \rightarrow 122.20$ for NCB in positive ion mode. $NAD^+$ and NMN were measured independently in this study, as their peaks were hardly separated within a short time.

sUA, cADPR, and FYU-981 (internal standard, 1 μM, Fuji Yakuhin Co., Ltd.) were eluted on CAPCELL PAK C8 TYPE UG 120 (2×150 mm, 5 μm) at 40°C. sUA and FYU-981 were separated using a gradient mobile phase containing water with 0.1% formic acid (A) and acetonitrile with 0.1% formic acid (B) at 0.4 mL/min. The elution was started with 55% A for 0.5 min, A was decreased from 55% to 5% for 1 min, then A was increased from 5% to 55% for 0.5 min, finally 55% A was maintained for 0.5 min. cADPR and FYU-981 were separated using an isocratic mobile phase containing 60% water with 0.1% formic acid and 40% acetonitrile with 0.1% formic acid at 0.4 mL/min. The selected transitions of m/z were $167.10 \rightarrow 124.10$ for sUA, $539.95 \rightarrow 272.95$ for cADPR, and $355.95 \rightarrow 159.90$ for FYU-981 in negative ion mode.

The injection volume was 1 μL for all measurements, and data manipulation was accomplished by Labsolutions software (Shimadzu).

## Statistical analysis

Statistics were performed using GraphPad Prism 9. Sample size was not predetermined by any statistical methods. Comparisons between multiple groups were performed using one-way ANOVA with Dunnett's or Tukey's multiple comparisons test, Kruskal-Wallis test, Brown-Forsythe and Welch ANOVA tests, or two-way ANOVA with Tukey's multiple comparisons test. Two-tailed unpaired t-test (with Welch's correction when applicable), or Mann-Whitney test were used for the analysis between two groups when applicable. Data were shown as mean ± s.e.m. or mean ± s.d. as indicated. $p < 0.05$ was considered significant.

## Acknowledgements

This study was supported by KAKENHI (JP21H02641) (IT), (JP23K18181) (IT), and (JP22K19372) (HA) from the Japan Society for the Promotion of Science (JSPS) and Research Grant 2022 (IT) from Gout and Uric Acid Foundation of Japan. SW was funded by the Japanese Government (Monbukagakusho: MEXT) Scholarship Program and Kanazawa University. The authors thank Dr. Zheng Jing, Ms. Aimi Taniguchi, Dr. Anpei Zhang, Mr. Kazuki Himi, and Mr. Kazuki Fujita for providing assistance.

## Additional information

### Funding

| Funder | Grant reference number | Author |
| --- | --- | --- |
| Japan Society for the Promotion of Science | KAKENHI [JP21H02641] | Ikumi Tamai |

| Funder | Grant reference number | Author |
| --- | --- | --- |
| Japan Society for the Promotion of Science | KAKENHI [JP23K18181] | Ikumi Tamai |
| Japan Society for the Promotion of Science | KAKENHI [JP22K19372] | Hiroshi Arakawa |
| Gout and Uric Acid Foundation | Research Grant 2022 | Ikumi Tamai |

The funders had no role in study design, data collection and interpretation, or the decision to submit the work for publication.

## Author contributions

Shijie Wen, Conceptualization, Data curation, Formal analysis, Validation, Investigation, Visualization, Methodology, Writing - original draft, Project administration, Writing – review and editing; Hiroshi Arakawa, Resources, Data curation, Formal analysis, Funding acquisition, Methodology, Project administration, Writing – review and editing; Shigeru Yokoyama, Yoshiyuki Shirasaka, Resources, Writing – review and editing; Haruhiro Higashida, Resources, Methodology, Writing – review and editing; Ikumi Tamai, Conceptualization, Resources, Data curation, Formal analysis, Supervision, Funding acquisition, Project administration, Writing – review and editing

## Author ORCIDs

Shijie Wen https://orcid.org/0000-0002-7686-5049
Ikumi Tamai https://orcid.org/0000-0003-4388-083X

## Ethics

All animal experiments were approved by the Institutional Animal Care and Use Committee at Kanazawa University (AP-214243).

Reviewer #1 (Public review): https://doi.org/10.7554/eLife.96962.3.sa1
Reviewer #2 (Public review): https://doi.org/10.7554/eLife.96962.3.sa2
Reviewer #3 (Public review): https://doi.org/10.7554/eLife.96962.3.sa3
Author response https://doi.org/10.7554/eLife.96962.3.sa4

# Additional files

## Supplementary files

• Supplementary file 1. Comparison between $K_i$ values and mean levels of soluble uric acid (sUA) in different tissues.

• Supplementary file 2. The data not shown in enzyme assays.

• Supplementary file 3. The raw images of chemical structures.

• MDAR checklist

## Data availability

No datasets or code are created or reused in this study. All data generated or analyzed in this study are provided in the manuscript text, figures, and supplementary files.

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
