## [Editor Report · eLife Assessment]

This **important** study shows that soluble uric acid is an endogenous inhibitor of CD38, a regulator of inflammatory responses. The **convincing** evidence draws both on biochemical analyses and in vivo models. This work provides insights into NAD+ metabolism, with significant implications for inflammation and potential roles in metabolic diseases and aging.

---

## [Referee Report · Reviewer #1 (Public review)]

This manuscript describes soluble Uric Acid (sUA) as an endogenous inhibitor of CD38, affecting CD38 activity and NAD+ levels both in vitro and in vivo. Importantly, the inhibition constants calculated supports the claim that sUA inhibits CD38 under physiological conditions. These findings are of extreme importance to understand the regulation of an enzyme that has been shown to be the main NAD+/NMN-degrading enzyme in mammals, which impacts several metabolic processes and has major implications to understanding aging diseases. The manuscript is well written, the figures are self explanatory, and in the experiments presented, the data is very solid. The authors discuss the main limitations of the study, especially in regard to the in vivo results. As a whole, I believe that this is a very interesting manuscript that will be appreciated by the scientific community and that opens a lot of new questions in the field of metabolism and aging.

During the revision process, the authors have performed new experiments to respond to relevant questions raised by the reviewers. In other cases, they have made changes in the text to improve the manuscript.

I believe that this manuscript in its current form is a mature and relevant set of findings that deserve attention and future developments.

---

## [Referee Report · Reviewer #2 (Public review)]

Summary:

This is an interesting work where Wen et al. aimed to shed light on the mechanisms driving the protective role of soluble uric acid (sUA) toward avoiding excessive inflammation. They present biochemical data to support that sUA inhibits the enzymatic activity of CD38 (Figures 1 and 2). In a mouse model of acute response to sUA and using mice deficient in CD38, they find evidence that sUA increases the plasma levels of nicotinamide nucleotides (NAD+ and NMN) (Figure 3) and that sUA reduces the plasma levels of inflammasome-driven cytokines IL-1b and IL-18 in response to endotoxin, both dependent on CD38 (Figure 4). Their work is an important advance in the understanding of the physiological role of sUA, with mechanistic insight that can have important clinical implications.

Strengths:

The authors present evidence from different approaches to support that sUA inhibits CD38, impacts NAD+ levels, and regulates inflammatory responses through CD38.

Weaknesses:

The authors investigate macrophages as the cells affected by sUA in promoting immunoregulation, proposing that sUA's inhibition of CD38 and the resulting increase in NAD+ promotes inflammasome inhibition through a previously established mechanism of NLRP3 regulation by NAD+-dependent sirtuins. However, they were unable to validate their in vivo findings using murine bone marrow-derived macrophages, a standard model for assessing inflammasome activation, due to the low uptake of sUA in these cells. Pharmacological blockage in THP-1 cells provides mechanistic evidence that sUA inhibits NLRP3-mediated secretion of IL-1β through CD38, but genetic evidence and direct assessment of the activation of inflammasome components would be necessary to fully validate the model.

---

## [Referee Report · Reviewer #3 (Public review)]

Summary:

In the present manuscript, the authors propose that soluble Uric acid (sUA) is an enzymatic inhibitor of the NADase CD38 and that it controls levels of NAD modulating inflammatory response. Although interesting the studies are at this stage preliminary and validation is needed.

Strengths:

The study characterizes the potential relevance of sUA in NAD metabolism.

Comment on revised version:

The authors have responded the majority of my criticism.

---

## [Author Response]

The following is the authors’ response to the original reviews.

**Public Reviews:**

**Reviewer #1 (Public Review):**
This manuscript describes soluble Uric Acid (sUA) as an endogenous inhibitor of CD38, affecting CD38 activity and NAD+ levels both in vitro and in vivo. Importantly, the inhibition constants calculated support the claim that sUA inhibits CD38 under physiological conditions. These findings are of extreme importance to understanding the regulation of an enzyme that has been shown to be the main NAD+/NMN-degrading enzyme in mammals, which impacts several metabolic processes and has major implications for understanding aging diseases. The manuscript is well written, the figures are self-explanatory, and in the experiments presented, the data is very solid. The authors discuss the main limitations of the study, especially in regard to the in vivo results. As a whole, I believe that this is a very interesting manuscript that will be appreciated by the scientific community and that opens a lot of new questions in the field of metabolism and aging. I found some issues that I believe constitute a weakness in the manuscript, and although they do not require new experiments, they may be considered by the authors for discussion in the final version of the manuscript.

We greatly appreciate the reviewer’s thoughtful comments and favorable review of our work.

The authors acknowledge the existence of several previous papers involving pharmacological inhibition of CD38 and their impact on several models of metabolism and aging. However, they only cite reviews. Given the focus of the manuscript, I believe that the seminal original papers should be cited.

Yes, we agreed with the reviewer. Two representative papers regarding the pioneering findings [Ref 1, 2] of pharmacological inhibition of CD38 were cited in the discussion of current manuscript.

(1) Tarragó, M. G., et al. (2018). A Potent and Specific CD38 Inhibitor Ameliorates Age-Related Metabolic Dysfunction by Reversing Tissue NAD+ Decline. Cell Metab 27(5): 1081-1095.e1010.

(2) Escande, C., et al. (2013). Flavonoid apigenin is an inhibitor of the NAD+ ase CD38: implications for cellular NAD+ metabolism, protein acetylation, and treatment of metabolic syndrome. Diabetes 62(4): 1084-1093.

Related to the previous comment, the authors show that they have identified the functional group on sUA that inhibits CD38, 1,3-dihydroimidazol-2-one. How does this group relate with previous structures that were shown to inhibit CD38 and do not have this chemical structure? Is sUA inhibiting CD38 in a different site? A crystallographic structure of CD38-78c is available in PDB that could be used to study or model these interactions.

Currently, there are several kinds of CD38 inhibitors, including NAD+/NMN analogs, flavonoids, 4-quinolines, etc. [Ref 1], but they do not have 1,3-dihydroimidazol-2-one or similar groups. We also noticed that sUA and its analogs have no remarkable structural similarity with these inhibitors. We have ever tried to identify the binding sites of sUA on CD38 by NMR. Since our NMR method required a large sample size, we had to prepare recombinant human CD38 using a cell-free protein synthesis system. However, the obtained CD38 protein showed a lower Vmax than commercial recombinant CD38 expressed in HEK293 cells, raising a concern of spatial conformation deference in the synthesized CD38. Thus, we were unable to get convinced data to confirm if sUA has different binding sites. Given the difference in structural feature and inhibition type, we did not use the PDB data regarding 78c-CD38 interaction for analysis in this study.

(1) Chini, E. N., et al. (2018). The Pharmacology of CD38/NADase: An Emerging Target in Cancer and Diseases of Aging. Trends Pharmacol Sci 39(4): 424-436.

Although the mouse model used to manipulate sUA levels is not ideal, the authors discuss its limitations, and importantly, they have CD38 KO mice as control. However, all the experiments were performed in very young mice, where CD38 expression is low in most tissues (10.1016/j.cmet.2016.05.006). This point should be mentioned in the discussion and maybe put in the context of variations of sUA levels during aging.

We appreciate the reviewer’s kind suggestions. Yes, CD38 expression in young mice is relatively low and we used young mice in this study; thus, aged mice would be promising to furthest evaluate the interaction between CD38 and sUA. Regarding the changes in sUA levels during aging, previous reports indicate that sUA levels seem to increase with age in mice and humans [Ref 1, 2]. We speculate that this increase is a physiologically compensatory response to aging in organisms. Accordingly, we added more details in the discussion (second paragraph).

(1) Iwama, M., et al. (2012). Uric acid levels in tissues and plasma of mice during aging. Biol Pharm Bull 35(8): 1367-1370.

(2) Kuzuya, M., et al. (2002). Effect of aging on serum uric acid levels: longitudinal changes in a large Japanese population group. J Gerontol A Biol Sci Med Sci 57(10): M660-664.

**Reviewer #2 (Public Review):**
Summary:This is an interesting work where Wen et al. aimed to shed light on the mechanisms driving the protective role of soluble uric acid (sUA) toward avoiding excessive inflammation. They present biochemical data to support that sUA inhibits the enzymatic activity of CD38 (Figures 1 and 2). In a mouse model of acute response to sUA and using mice deficient in CD38, they find evidence that sUA increases the plasma levels of nicotinamide nucleotides (NAD+ and NMN) (Figure 3) and that sUA reduces the plasma levels of inflammasome-driven cytokines IL-1b and IL-18 in response to endotoxin, both dependent on CD38 (Figure 4). Their work is an important advance in the understanding of the physiological role of sUA, with mechanistic insight that can have important clinical implications.Strengths:The authors present evidence from different approaches to support that sUA inhibits CD38, impacts NAD+ levels, and regulates inflammatory responses through CD38.

We deeply thank the reviewer for the thoughtful comments and appreciation of our findings.

Weaknesses:The authors investigate macrophages as the cells impacted by sUA to promote immunoregulation, proposing that inflammasome inhibition occurs through NAD+ accumulation and sirtuin activity due to sUA inhibition of CD38. Unfortunately, the study still lacks data to support this model, as they could not replicate their in vivo findings using murine bone marrow-derived macrophages, a standard model to assess inflammasome activation. Without an alternative approach, the study lacks data to establish in vitro that sUA inhibition of CD38 reduces inflammasome activation in macrophages - consequently, they cannot determine yet if both NAD+ accumulation and sirtuin activity in macrophages is a mechanism leading to sUA role in vivo.

We deeply thank the reviewer for pointing out this weakness in our work. In fact, we tried to prepare stable CD38 KD/KO THP-1 cells in the middle of 2021; however, we faced some technical problems due to the limitations of instruments. Thus, we used CD38 KO mice to prepare CD38 KO BMDMs, as shown in the first version of manuscript, we failed to replicate the results in BMDMs because of the low uptake of sUA. To address the reviewer’s concern regarding the lack of an in vitro link between CD38 and sUA immunosuppression, we used 78c, a highly specific and potent inhibitor of CD38, to block CD38 in primed THP-1 cells. Then we evaluated the effect of sUA pre-incubation on MSU crystal-induced IL-1β release in primed THP-1 cells (vehicle and CD38 blockade). The added results in Figure 4-figure supplement 2B and 2C indicated that CD38 blockade largely impaired the immunosuppressive effect of sUA without reducing sUA uptake. In addition, we found that sUA at physiological levels boosted NAD+ levels in THP-1 cells (Figure 3-figure supplement 1B) without affecting the activities of other key enzymes involved in NAD+ synthesis and degradation, including NAMPT and PARP (Figure 3-figure supplement 2). All these results supported that CD38 is a key mediator for sUA at physiological levels to regulate inflammasome activation in vitro.

**Reviewer #3 (Public Review):**
Summary:In the present manuscript, the authors propose that soluble Uric acid (sUA) is an enzymatic inhibitor of the NADase CD38 and that it controls levels of NAD modulating inflammatory response. Although interesting the studies are at this stage preliminary and validation is needed.Strengths:The study characterizes the potential relevance of sUA in NAD metabolism.

We greatly appreciate the reviewer for the thoughtful comments and valuable suggestions.

Weaknesses:(1) A full characterization of the effect of sUA in other NAD-consuming and synthesizing enzymes is needed to validate the statement that the mechanism of regulation of NAD by sUA is mediated by CD38, The CD38 KO may not serve as the ideal control since it may saturate NAD levels already. Analysis of multiple tissues is needed.

Yes, it is necessary to confirm if sUA affects other NAD+-consuming and synthesizing enzymes. To address the concern and to provide additional validation, we tested the direct effects of sUA and other purine derivates on the activities of another two key enzymes involved in the metabolic network of NAD+, including PARP (NAD+-consuming enzyme) and NAMPT (NAD+-synthesizing enzyme). The added results in Figure 3-figure supplement 2 showed that sUA has no effect on PARP and NAMPT activity, suggesting that CD38 is a main target for sUA in regulating NAD+ availability. In addition, we also confirmed both PARP and NAMPT were not affected by purine metabolism under physiological conditions. Although hypoxanthine and xanthine, at 500 μM (supraphysiological levels), slightly inhibited PARP activity, it has no physiological significance due to their low physiological concentrations (generally below 20 μM). Further evaluation of these inhibitory effects under pathological conditions would be of interest but were beyond the focus of this study.

Given that tissue sUA uptake is saturated under physiological conditions (tissue sUA did not increase in our models, Figure 3-figure supplement 5A and 5B), CD38 and other potential targets in tissues may be not affected by sUA in our models. We used CD38 KO mice to confirm if sUA interacts with other targets to regulate NAD+ degradation and inflammatory responses. A previous study [Ref 1] revealed that inhibition of other enzymes involved in NAD+ metabolism, such as PARP, resulted in a significant increase of NAD+ availability in CD38 KO mice, which indicates that CD38 KO mice can be used to exclude the potential interaction between sUA and other targets. In fact, we did not observe significant effects of sUA in CD38 KO mice. More importantly, we added the additional validation regarding PARP and NAMPT activity according to the reviewer’s kind suggestion, which further confirmed that CD38 is the main target for sUA in our models.

(1) Tarragó, M. G., et al. (2018). A Potent and Specific CD38 Inhibitor Ameliorates Age-Related Metabolic Dysfunction by Reversing Tissue NAD+ Decline. Cell Metab 27(5): 1081-1095.e1010.

(2) The physiological role of sUA as an endogenous inhibitor of CD38 needs stronger validation (sUA deficient model?).

We thank the reviewer’s insightful suggestions. Yes, sUA depletion model is ideal for further validation, as we discussed in the limitations of this study. Given that introduction of exogenous recombinant uricase (immunometabolism may be affected) to deplete sUA is not ideal for the evaluation under physiological conditions, uricase-transgenic mice would be a promising model. However, now we have no uricase-transgenic mice, and we are unable to prepare CD38 KO/uricase-transgenic mice for additional validation within a reasonable time. In the first version of manuscript, therefore, we used an sUA-release model in sUA-supplementation mice as a further validation in Figure 3E.

(3) Flux studies would also be necessary to make the conclusion stronger.

Answer: We highly appreciate the reviewer’s suggestion regarding metabolic flux analysis. Yes, flux analysis using specifically designed isotope-labeled NAD+ is an ideal validation in mice, as it can track any sUA-induced changes in NAD+ metabolism. However, we are unable to synthesize or obtain suitable isotope-labeled substrates for in vivo validation due to the technical limitations and financial burdens.

**Recommendations for the authors:**

**Reviewer #1 (Recommendations For The Authors):**
The manuscript is very solid and very well-presented and discussed. In my opinion, the only weakness in the writing is that the message about finding an endogenous regulator of CD38 activity and NAD levels gets blurred by the temptation of jumping into the potential of developing new pharmacological CD38 inhibitors based on sUA structure. My recommendation would be to focus on delivering a clear message about sUA as a physiological inhibitor of CD38, and the possible implications for understanding the onset and evolution of metabolic diseases and aging. Maybe leave the potential of developing novel sUA-based CD38 inhibitors for a final comment. I understand this last point is very attractive, but there are very potent pharmacological CD38 inhibitors already available with promising results.

We greatly appreciate the reviewer’s valuable suggestions. Yes, there are some promising CD38 inhibitors with nanomolar Ki such as 78c. To clearly focus on sUA as a physiological inhibitor of CD38, we simplified the description in the manuscript and just keep the discussion of functional group. Since the data regarding the development of CD38 inhibitors in our manuscript remain limited, we did not put it in a separate part. We believe the simplified information is still sufficient for medicinal chemists who are interested in the development of CD38 inhibitors.

**Reviewer #2 (Recommendations For The Authors):**
Major comments:(1) The authors present several pieces of data to explain why there is no impact of sUA in inflammasome-mediated cytokine secretion, which is convincing and supports that BMDM will not be a model of choice to establish macrophages as the target of sUA. However, their data with THP-1 cells is compelling and could be further explored using shRNA, or CRISPR approaches to deplete CD38, thus establishing a mechanistic link in vitro.

We greatly appreciate the reviewer’s thoughtful comments. We added more results as mentioned before (please see our response to public review).

(2) There is a severe lack of linearity in how the figures are presented in the main text, making it difficult to read through the manuscript. The figures should be presented in the order they appear in the panels.

We thank the reviewer for pointing out this issue. We improved the figure assembly according to eLife guidelines.

Minor comments:(1) The authors do not appropriately address the finding that CD38 impacts the secretion of IL-1b in BMDM (Figure S6B) and in vivo (Figure 4), possibly independently of sUA.

Yes, the regulatory effect of CD38 on cytokine release seems independent of sUA. In fact, we used CD38 KO BMDMs to validate the role of CD38 in the inflammasome activation. We showed that sUA levels are comparable between WT and KO mice, suggesting that CD38 KO does not affect the baselines and boosted levels of sUA in our models. In this situation, we were able to evaluate the immunosuppressive effects of sUA at the same physiological levels in WT and CD38 KO mice, thus providing evidence to support that sUA at physiological levels limits excessive inflammation via CD38.

(2) Figure 3F: the legend on the x-axis lacks the indication of which groups were treated with recombinant hCD38.

We appreciate the reviewer’s comments. We improved Figure 3 by adding more information.

(3) While the results on panels 3 and 4 provide robust evidence that sUA is anti-inflammatory through CD38, the title of the figures extrapolates their findings (i.e., no data shows CD38-sUA, either in vitro or in vivo).

We appreciate the reviewer’s kind suggestion. We provided data to support the direct interaction between CD38 and sUA in Figure 3F; we admitted that we did not show the data regarding the direct interaction in mice in Figure 4. To help readers easily track the results, however, we used a conclusion-like title.

(4) The introduction could briefly mention NMN and CD38 activity as an ecto-enzyme to facilitate the understanding of their findings by a general audience, especially the dosing of NMN and their data on BMDM.

We added more description regarding CD38 and NMN in the introduction part. Once again, we deeply thank the reviewer for the valuable suggestions.

**Reviewer #3 (Recommendations For The Authors):**
(1) A full characterization of the effect of sUA in other NAD-consuming and synthesizing enzymes is needed to validate the statement that the mechanism of regulation of NAD by sUA is mediated by CD38, The CD38 KO may not serve as the ideal control since it may saturate NAD levels already. Analysis of multiple tissues is needed.

We greatly appreciate the reviewer’s valuable suggestions. We added more results to validate CD38 as the main target of sUA in our models. Please see our response to public review.

(2) The physiological role of sUA as an endogenous inhibitor of CD38 needs stronger validation (sUA deficient model?).

We greatly appreciate the reviewer’s valuable suggestions. Please see our response to public review.

(3) Flux studies would also be necessary to make the conclusion stronger.

We greatly appreciate the reviewer’s valuable suggestions. Please see our response to public review.